# Genome-Wide Identification, Characterization and Experimental Expression Analysis of CNGC Gene Family in *Gossypium*

**DOI:** 10.3390/ijms24054617

**Published:** 2023-02-27

**Authors:** Lei Chen, Wenwen Wang, Hailun He, Peng Yang, Xiaoting Sun, Zhengsheng Zhang

**Affiliations:** Engineering Research Center of South Upland Agriculture, Southwest University, Ministry of Education, Chongqing 400716, China

**Keywords:** *Gossypium*, CNGC, phylogenetic analysis, oxidative stress, hormones

## Abstract

Cyclic nucleotide-gated ion channels (CNGCs) are channel proteins for calcium ions, and have been reported to play important roles in regulating survival and environmental response of various plants. However, little is known about how the CNGC family works in *Gossypium*. In this study, 173 CNGC genes, which were identified from two diploid and five tetraploid *Gossypium* species, were classified into four groups by phylogenetic analysis. The collinearity results demonstrated that CNGC genes are integrally conservative among *Gossypium* species, but four gene losses and three simple translocations were detected, which is beneficial to analyzing the evolution of CNGCs in *Gossypium*. The various cis-acting regulatory elements in the CNGCs’ upstream sequences revealed their possible functions in responding to multiple stimuli such as hormonal changes and abiotic stresses. In addition, expression levels of 14 CNGC genes changed significantly after being treated with various hormones. The findings in this study will contribute to understanding the function of the CNGC family in cotton, and lay a foundation for unraveling the molecular mechanism of cotton plants’ response to hormonal changes.

## 1. Introduction

Cyclic nucleotide gated channel (CNGC) protein is one of the calcium ion conduction pathways and widely exists in animals and plants. Plant CNGCs are members of the “P-loop” superfamily of cation channels, which is found in all prokaryotic and eukaryotic cells [1]. P-loop channels, which emerged early in evolution, have evolved into channels that conduct various cations and serve a wide range of functions in cells [2]. The calcium ion (Ca^2+^) is an important secondary messenger that modulates multiple signaling pathways in plants. Calcium signal transduction is the main mechanism for plants to receive and respond to endogenous and exogenetic stimuli, such as hormones [3], pathogens [4], salt stress [5], optical signals [6] and circadian rhythm [7]. As cation channel proteins, CNGCs have been reported to mediate the accumulation of Ca^2+^ in cytosol and the conversion of external optical and odor signals into bioelectrical signals [8,9]. For example, *AtCNGC1* contributes to Ca^2+^ uptake into plants [10], while *AtCNGC3* functions as a Na^+^ uptake and a K^+^ uptake mechanism and is required for cellular homeostasis [11]. Furthermore, CNGCs are also involved in plant development and disease resistance [12,13]. Silencing of both *TaCNGC14* and *TaCNGC16* enhances wheat resistance against Pst [14].

With the identification of the first *CNGC* (*HVCBT1*) in barley, a series of gene family members have been identified in plants in recent years, such as *Arabidopsis*, rice, pear, tobacco, Chinese cabbage, *Brassica napus*, *Brassica oleracea*, tomato, maize and wheat [14,15,16,17,18,19,20,21,22,23]. Based on the gene structure, the CNGC family could be classified into four groups (group I–IV), and group IV can be further subdivided into “IV-A and IV-B” [15]. It was reported that different groups of tomato CNGC genes play distinguishable roles in disease resistance and abiotic stress responses [22]. In plants, the CNGC consists of six transmembrane (TM) domains and a pore region located in the middle of the fifth and sixth TM domains. The cyclic nucleotide binding domain (CNBD) is a highly conserved domain with a phosphate binding cassette (PBC) and a “hinge” region [22]. A specific motif of plant CNGC ([LIMV0]-X(2)-[GSANCR]-X-[FVIYASCL]-XGX(0,1)-X(0,1)-[EDAQGH]-L-[LIVFA]-X-[WRCMLS0]-X-[LMSIQAFT0]-X(7,37)-[SAC]-X(9)-[VTIALMS]-X(0,1)-[EQDN]-[AGSVT]-[FYL]-X-[LIVF]) located in the PBC and “hinge” region of CNBD and existed uniquely in the plant CNGC, and this motif has been treated as a criterion to identify the CNGC gene family in plants [22,24].

CNGCs are involved in different signaling pathways associated with the response to biotic and abiotic stress, including salt, drought, cold, plant nutrition and calcium homeostasis and pathogen infection. In this study, CNGC families of *Gossypium* species were identified. The chromosome location and synteny among species of CNGCs, along with their gene structure, domain composition and cis-acting regulatory elements, were further analyzed. Moreover, the expression levels of *Gossypium* CNGC genes among tissues and in response to hormones and oxidative stress were quantitated to understand their potential functions. These findings will be beneficial to understanding the function of the CNGC family in cotton, and lay a foundation for unraveling the molecular mechanism of cotton plants’ response to hormonal changes.

## 2. Results

### 2.1. CNGC Identification in Seven Gossypium Species

A total of 17, 14, 28, 28, 29, 28 and 29 genes were identified in *Gossypium arboreum*, *Gossypium raimondii*, *Gossypium barbadense*, *Gossypium darwinii*, *Gossypium hirsutum*, *Gossypium tomentosum* and *Gossypium mustelinum*, respectively. Based on their chromosomal location, CNGC genes were named *GaCNGC1* to *GaCNGC17* in *G. arboreum*, *GbCNGC1* to *GbCNGC28* in *G. barbadense*, *GdCNGC1* to *GdCNGC28* in *G. darwinii*, *GhCNGC1* to *GhCNGC29* in *G. hirsutum*, *GrCNGC1* to *GrCNGC14* in *G. raimondii*, *GtCNGC1* to *GtCNGC28* in *G. tomentosum* and *GmCNGC1* to *GmCNGC29* in *G. mustelinum*. The protein size of CNGCs ranged from 517 (aa) to 769 (aa). The theoretical isoelectric point (PI) of most CNGC proteins was greater than eight, with the exception of six *Gossypium* CNGC genes (*GbCNGC19*, *GdCNGC19*, *GhCNGC19*, *GmCNGC20*, *GrCNGC12*, *GtCNGC19*) that have a theoretical PI of 6.53 and five *Gossypium* CNGC genes (*GaCNGC4*, *GbCNGC6*, *GdCNGC6*, *GmCNGC7* and *GtCNGC6*) with a theoretical PI of 7.52 (Appendix A). These findings indicated that the number of CNGCs varies slightly between species, which may have been caused by the loss or duplication of genes during the evolution of *Gossypium*. Furthermore, the *Gossypium* CNGC proteins are a good source of fundamental amino acids.

### 2.2. Phylogenetic Analysis of the CNGC Family

In order to investigate the genetic evolutionary relationship of CNGCs, a neighbor-joining tree was constructed using the maximum likelihood (ML) method with a total of 209 CNGC proteins, including 20 from *Arabidopsis* [15], 16 from rice [16] and 173 from *Gossypium* species. (Figure 1). As a result, these CNGCs were unevenly classified into four groups, and group IV was further divided into subgroup IVa and subgroup IVb. Group I included four GaCNGCs, six GbCNGCs, six GdCNGCs, six GhCNGCs, six GmCNGCs, three GrCNGCs, six GtCNGCs, six AtCNGCs and three OsCNGCs. Group II included three GaCNGCs, four GbCNGCs, five GdCNGCs, five GhCNGCs, six GmCNGCs, three GrCNGCs, five GtCNGCs, five AtCNGCs and three OsCNGCs. Group III contained 6 GaCNGCs, 12 GbCNGCs, 12 GdCNGCs, 12 GhCNGCs, 12 GmCNGCs, 6 GrCNGCs, 12 GtCNGCs, 5 AtCNGCs and 5 OsCNGCs. Group III has 82 CNGCs, which is the largest branch of the four CNGC groups. Group IVb contained four GaCNGCs, five GbCNGCs, five GdCNGCs, six GhCNGCs, five GmCNGCs, two GrCNGCs, five GtCNGCs, two AtCNGCs and three OsCNGCs. However, no CNGCs from Gossypium were contained in group IVa, indicating the peculiarity of cotton species.

### 2.3. Chromosomal Distribution and Synteny Analysis of Gossypium CNGCs

The chromosomal distribution of *Gossypium* CNGC genes is shown in Figure 2 and Appendix A. For diploid species, *G. arboreum* and *G. raimondii*, CNGC genes were located on eight chromosomes. For five allotetraploid species, CNGC genes were located on 17 chromosomes in *G. hirsutum*, and spread on 16 chromosomes in the other four allotetraploid species. CNGCs are unevenly distributed on chromosomes in *Gossypium*. In allotetraploid species, most chromosomes contained only one CNGC gene, while chromosomes A_t_05 and D_t_04 contained six and five CNGCs, respectively.

The synteny analysis of *Gossypium* CNGCs showed that the CNGC genes were well distributed on the homologous chromosome except for the CNGC gene on the reciprocal translocation chromosomes At4 and At5 in allotetraploid species. Meanwhile, three simple translocations and two deletions existed in *Gossypium*. The CNGC gene located on chromosome A_t_7 or D_t_7 of allotetraploid species was simply translocated on chromosome 4 of *G*. *arboretum*, and one CNGC gene located on chromosome A_t_3 or D_t_3 of allotetraploid species was simply translocated on chromosome 2 of *G. arboretum*. One CNGC gene located on chromosome A_t_13 or D_t_13 of allotetraploid species was simply translocated on chromosome 10 of *G. raimondii*. The CNGC gene existed in chromosome A_t_9 of allotetraploids, but it was absent in the D_t_ subgenome and *G. raimondii*. The CNGC gene occurred on chromosomes A_t_6 and D_t_6 of tetraploid species or on chromosome 6 of *G*. *arboretum*, but it was absent in *G. raimondii*.

### 2.4. Genic Structure and Motif Composition Analysis

To further reveal the phylogenetic relationships of CNGCs, the gene structure of CNGCs was visualized by TBtools based on genome annotation files from each species (Figure 3). The results showed that *Gossypium CNGCs* had slightly different but similar structural characteristics. The majority of *Gossypium* CNGC genes contained seven exons. In addition, there were 13 genes (*GaCNGC4*, *GbCNGC6*, *GtCNGC6*, *GdCNGC6*, *GmCNGC7*, *GaCNGC4*, *GhCNGC6*, *GrCNGC12*, *GhCNGC19*, *GbCNGC19*, *GmCNGC20*, *GdCNGC19*, *GtCNGC19*) and 9 genes (*GaCNGC11*, *GaCNGC12*, *GbCNGC12*, *GdCNGC12*, *GmCNGC12*, *GhCNGC12*, *GtCNGC12*, *GaCNGC14*, *GhCNGC18*) that had 6 and 8 exons, respectively. Additionally, the last exons of genes in group III were longer than all the other exons. The similar genic structure of CNGC genes indicated their evolutionary conservation [25]. However, there were a few CNGC members with different genic structures, which suggested that independent evolutionary events occurred in them.

A total of 15 conserved motifs, labeled motif 1 to motif 15, were identified in *Gossypium* CNGCs (Figure 3). Motif 5, which belongs to the cyclic nucleotide-binding domain (CNBD), was the conserved domain contained in all the *Gossypium* CNGC genes. The majority of CNGCs contained all 15 motifs, with different arrangements in each group. However, there were a few CNGCs with fewer than 15 motifs, indicating that motif deletions occurred in them during evolution. *GmCNGC19*, with only 10 motifs, was the one with the fewest motifs. These findings demonstrated that most CNGC genes were conserved, with a few exceptions being due to evolutionary variation.

### 2.5. Domain Composition and Conserved Motif at the PBC and Hinge Region of CNB Domain of CNGC Proteins in Plants

Plant CNGCs are composed of three main domains including the TM /ITP domain, CNB domain and calmodulin-binding (CaMB) domain, with the latter two domains usually overlapping partially [12]. The CNB domain ([LI]-X(2)-[GS]-X-[FYIVS]-X-G-X(0,1)-[DE]-LL-X(8,25)-[SA]-X(9)-[VLIT]-E-X-F-X-[IL]), which contained a PBC region and a “hinge” region, was the distinguishing feature of the CNGC protein [24]. To investigate the CNB domain of *Gossypium* CNGCs, protein sequences of the PBC and “hinge” regions from rice and *Arabidopsis* were used as controls, and the PBC and “hinge” region of 173 CNGC proteins in cotton were then aligned. As a result, individual CNB motifs for each group were created as follows: [LI]-X-A-G-D-F-C-G-[ED]-X-LL-X(25)-E-A-F-A-L for group I, L-K-E-G-D-F-C-G-E-E-LL-X(25)-E-A-F-A-L for group II, [LI]-X-P-G-D-F-C-G-E-E-LL-X(25)-E-A-F-A-L for group III and [LI]-X(2)-G-X-F-X-G-D-E-LL-X(25)-E-A-F-G-L for group IVb (Figure 4).

### 2.6. Prediction of Cis-Acting Regulatory Elements

To determine the preliminary function of the GhCNGCs, 2000 bp sequences in the upstream region were cropped in order to analyze the cis-acting elements in the potential promoter regions. The results revealed that the cis-acting elements of the GhCNGC genes from the same clade were similar, which was consistent with their genetic structure and phylogenetic relationships (Figure 5). A variety of cis-elements of *GhCNGCs* were related to various exogenous stimuli, such as abscisic acid responsiveness elements, anaerobic induction elements, auxin-responsive elements, defense and stress responsiveness elements, gibberellin-responsive elements, MeJA-responsive elements and salicylic acid responsiveness elements. These elements, which are crucial for plants to respond abiotic stress [26], suggested that abiotic stressors might regulate the expression of GhCNGCs.

### 2.7. Expression Profiling of CNGC Family in G. hirsutum

To explore the biological functions of the CNGC family, the transcriptome of *G. hirsutum* was analyzed to investigate the expression profiles of GhCNGCs in various tissues, including fiber, calycle, leaf, petal, pistil, root, stamen, stem, torus and ovule (Figure 6). The results revealed that most CNGCs were expressed in almost all tissues, but the dominantly expressed tissues were diverse for different genes. In detail, *GhCNGC12* and *GhCNGC18* were highly expressed in calycle, petal, stamen and ovule. *GhCNGC14* and *GhCNGC29* were highly expressed in the stem and torus. *GhCNGC6* and *GhCNGC19* were highly expressed in the stamen. *GhCNGC14* and *GhCNGC10* were highly expressed in the petal. Additionally, 16 GhCNGC genes were highly expressed in the leaf. These findings suggested that the main functional tissues of different GhCNGCs might be distinct.

### 2.8. Response of GhCNGCs to Oxidative Stress and Hormones

To investigate whether *GhCNGCs* were regulated by oxidative stress or hormones, the expression levels of *GhCNGCs* were analyzed using qRT-PCR. For *GhCNGCs* distributed on homologous chromosomes, only the one on the At subgenome was analyzed. In total, the expression levels of 15 *GhCNGCs* were qualified, and 14 of them, with the exception for *GhCNGC6*, showed significant expression changes in response to various stimuli (Figure 7). *GhCNGC6* was not expressed in either the CK or treatment assays. Most of the other *GhCNGCs* were significantly upregulated in response to exogenous stimuli. For instance, the IAA, SA, MeJA, GA, ABA, ETH and H_2_O_2_ treatments significantly upregulated 2, 8, 6, 13, 11, 10 and 2 *GhCNGCs*, respectively. Additionally, five, four, two, one and seven *GhCNGCs* were significantly downregulated by the IAA, SA, MeJA, ETH and H_2_O_2_ treatments, respectively. Furthermore, *GhCNGC10* was significantly regulated by all the treatments. These findings supported the hypothesis that *CNGCs* in cotton respond to changes in oxidative stress and hormone levels.

## 3. Discussion

In this study, 173 CNGC genes from seven *Gossypium* species, together with 20 *AtCNGCs* [15] and 16 *OsCNGCs* [16], from a total of 209 *CNGCs* were used to construct a phylogenetic tree, and the CNGC genes were classed into five groupings after examination. However, group IVa CNGC genes did not exist in *Gossypium*, and it was similar to maize [22]. This finding supported that group IVa CNGC genes were more ancient in the evolution of green plants [22]. In addition, four gene losses and three simple translocations were identified in seven *Gossypium* species. The first gene loss was found on chromosome A_t_05 of allotetraploid *Gossypium* species, but this gene could be found in two diploid species and the D_t_ subgenome of tetraploid species. This finding indicated that the gene loss on chromosome A_t_05 of allotetraploid species occurred during the formation of the reciprocal translocation on chromosomes At4 and At5 in allotetraploid species. The second gene loss was found on chromosome D_t_09 of allotetraploid species and *G. raimondii*, but this gene could be found on chromosome A_t_9 in allotetraploid species. The gene loss on chromosome D_t_9 in allotetraploid species occurred in *G. raimondii* before the merging of the two diploid species.

In the previous study, CNGC genes were identified and characterized in several species, including 20, 16, 21, 35, 30, 61, 26, 18, 12, 16, 16 and 14 CNGC genes in *Arabidopsis*, rice, pear, tobacco, Chinese cabbage, *Brassica napus*, *Brassica oleracea*, tomato, maize and subgenomes A, B, D of wheat, respectively [14,15,16,17,18,19,20,21,22,23]. In the present study, 17 and 14 CNGC genes were identified in diploid *Gossypium* species *G. arboretum* and *G. raimondii*, respectively. The number of CNGC genes in the A_t_ subgenome or D_t_ subgenome was also inconstant compared with other species. The difference in CNGC number might have resulted from the evolution and domestication of different species. In addition, allotetraploid *Gossypium* was formed after hybridization between the A genome and D genome and chromosome doubling [27]. Theoretically, the number of CNGC genes in allotetraploid species was double that of diploid species. However, all allotetraploid species had fewer CNGC genes than the sum of the two diploid *Gossypium* species. This phenomenon might be due to the ongoing process of gene loss in allotetraploid species [28].

Sequence alignment analysis showed that CNGC genes possessed a highly conserved motif. In this study, 173 *Gossypium* CNGC genes from 2 diploid and 5 allotetraploid *Gossypium* species were classified into 4 groups (I-IV). The motifs for the four groups were [LI]-X-A-G-D-F-C-G-[ED]-X-LL-X(25)-E-A-F-A-L, L-K-E-G-D-F-C-G-E-E-LL-X(25)-E-A-F-A-L, [LI]-X-P-G-D-F-C-G-E-E-LL-X(25)-E-A-F-A-L and [LI]-X(2)-G-X-F-X-G-D-E-LL-X(25)-E-A-F-G-L, respectively. Subsequently, the sequence structure of *Gossypium*, [LI]-X(2)-G-X-F-X-G-[ED]-X-L-L-X(25)-E-A-F-[AG]-L, was obtained. Additionally, the conserved motif of *Gossypium* was similar to other species with a slight difference, such as wheat ([LI]-X(2)-[GS]-X-[FCV]-X-G-[ED]-E-L-L-[TGS]-W-X-[LF]-X(7,17)-[LFR]-[PL]-X-[SA]-X(2)-[TS]-X(6)-[VAT]-[EQ]-X-F-X-L-X-[AS]-X-[DE]-[LV]) and rice ([LI]-X(2)-[GS]-X-[FV]-X-G-[DE]-ELL-X-W-X(12,22)-SA-X(2)-T-X(7)-[EQ]-AF-X-L) [14,16]. The CNGC genes from different species had a highly conserved sequence, so they might possess similar functions.

The previous study showed that the CNGC genes were involved in biotic and abiotic stress in plants, for example, the expression of *BrCNGC7* was downregulated under temperature stress but upregulated under salt and osmotic stress [19]. *OsCNGC6* responds to stress and plant defense-related functions in rice [16]. *BoCNGC* genes from phylogenetic groups I and IV were particularly sensitive to cold stress and infections with bacterial pathogens [21]. Four genes (*BnaC03g31050D*, *BnaC03g31720D*, *BnaA05g01380D* and *BnaC04g01250D*) from group I and two genes (*BnaCnng45430D* and *BnaA03g34680D*) from group IVa were all strongly induced by SA and infection of *S. sclerotiorum* [20]. VIGS analysis demonstrates that *SlCNGC17* and *SlCNGC18* play a role in resistance to *P. aphanidermatum* [22]. *TaCNGC14* and *TaCNGC16* play a negative role in wheat resistance against pathogens [29]. Notable performances were exhibited by group I and IV *NtabCNGC* genes against black shank [18]. In this study, the response of *GhCNGC* genes to hormone treatment indicated that the *GhCNGC* genes were regulated by several hormones, for example, twelve *GhCNGC* genes were induced by SA treatment. The previous study showed that SA plays crucial roles in plant defense against pathogens [30]. In addition, *AtCNGC2* and *AtCNGC6* have been reported as having a key role in stress response [31,32,33]. In this study, *GhCNGC2* and *GhCNGC7/18* were identified as homologous genes of *AtCNGC2*. *GhCNGC4/23*, *GhCNGC14/29* and *GhCNGC21* were homologous genes of *AtCNGC6*. These findings suggest the putative roles of these *GhCNGC* genes may be involved in biotic and abiotic stress through hormone signaling pathways.

## 4. Materials and Methods

### 4.1. Plant Growth and Treatments

CCRI 35, a *G. hirsutum* cultivar, was planted in potting soil at 25 °C in a culture room with a 16 h light/8 h dark cycle to determine the responses of GhCNGCs to oxidative stress and hormones. As treatment assays, abscisic acid (ABA, 100 μM), salicylic acid (SA, 1 mM), methyl jasmonate (MeJA, 100 μM), auxin (IAA, 5 μM), gibberellin (GA, 0.5 μM), ethylene (ETH, 200 μM) and H_2_O_2_ (100 mM) were sprayed individually onto cotton leaves. The control assay was performed with pure water. To prepare these treating solutions, the corresponding powder was dissolved to the desired concentration in deionized water. For powder that was difficult to dissolve in water, ethanol (ABA, IAA, SA) or methanol (GA) was added as a hydrotropic agent. After being treated for 10 h, leaf samples were flash frozen by liquid nitrogen and then stored at −80 °C until further experiments were performed.

### 4.2. Gene Identification and Characterization Analysis

Genome sequences of seven *Gossypium* species, including *G. arboreum*, *G. raimondii*, *G. barbadense*, *G. darwinii*, *G. hirsutum*, *G. tomentosum* and *G. mustelinum*, were downloaded from NCBI (https://www.ncbi.nlm.nih.gov/, accessed on 1 September 2022). The protein sequences of 20 *Arabidopsis* CNGCs and 16 rice CNGCs were downloaded from the *Arabidopsis* Information Resource (TAIR10) database (http://www.arabidopsis.org/) (accessed on 5 September 2022) and Rice Genome Annotation Project (RGAP) database (http://rice.plantbiology.msu.edu/) (accessed on 7 September 2022), respectively. AtCNGC amino acid sequences were used as queries in local Basic Local Alignment Search Tool protein (BLASTP) [34] (e-value of 1 × 10^−10^) searches to identify potential CNGC proteins in seven *Gossypium* species. In addition, the HMM profiles of the CNB domain (PF00027) from the Pfam database (http://pfam.xfam.org, [35] (accessed on 7 September 2022)) was used to search the cotton protein database using HMMER 3.0 (http://hmmer.janelia.org/, accessed on 8 September 2022) [36]. The intersection of results from BLASTP and HMMER was selected. In addition, the Simple Modular Architecture Research Tool (SMART) (http://smart.embl-heidelberg.de/, accessed on 11 September 2022) [37] and InterPro (http://www.ebi.ac.uk/interpro/, accessed on 11 September 2022) [38] were utilized to confirm the domains of selected proteins.

### 4.3. Analysis of Gossypium CNGC Protein Features and Chromosomal Localization

Protein features, including the number of amino acids, molecular weight, theoretical isoelectric point (PI), instability index, aliphatic index and grand average of hydropathicity (GRAVY), were analyzed by ExPASy (https://web.expasy.org/protparam/, accessed on 9 September 2022) [29]. Additionally, locations of CNGCs were obtained from genomic annotation. TBtools [39] was used to display the locations of genes on chromosomes. In addition, JCVI (https://www.cnpython.com/pypi/jcvi, accessed on 10 September 2022) was used to display the synteny of *Gossypium* CNGCs in seven different cotton species.

### 4.4. Analysis of Protein Motifs, Gene Structures and Cis-Acting Regulatory Elements

Protein sequences of the CNGCs were uploaded to MEME (https://meme-suite.org/meme/tools/meme, accessed on 12 September 2022) to identify conserved motifs with the parameters of 15 motifs. The motifs of *Gossypium* CNGCs were displayed by TBtools [39].

For promoter analysis, 2000 bp sequences at the upstream of the start codon (ATG) of *GhCNGC* genes were obtained as potential promoter regions. PlantCARE was used to search promoter regions for cis-acting regulatory elements. The cis-acting regulatory elements in the promoter regions of CNGC genes were classified on the basis of their responses to different hormones and abiotic factors.

### 4.5. Phylogenetic Analysis and Multiple Sequence Alignment

CNGC protein sequences from *Arabidopsis thaliana*, *Oryza sativa*, *G. arboreum*, *G. barbadense*, *G. darwinii*, *G. hirsutum*, *G. raimondii*, *G. tomentosum* and *G. mustelinum* were aligned using the MUSCLE program in MEGA 7.0 with default settings [40]. The alignment was further used to construct the neighbor-joining (NJ) tree with the following parameters: 1000 bootstrap replicates, Poisson model, uniform rates and pairwise deletion. The conserved domains were aligned and shaded with DNAMAN software.

### 4.6. Transcriptome Analysis of CNGC Genes in Different Tissues

RNA-seq data of the *G. hirsutum* cultivar TM-1 [28] were downloaded from NCBI (http://www.ncbi.nlm.nih.gov/, accessed on 1 September 2022). These data were used to analyze the expression of *GhCNGC* genes in eight different tissues (calycle, leaf, petal, pistil, root, stamen, stem, torus) and at different fiber and ovule developmental stages.

### 4.7. RNA Isolation and qPCR

Total RNA was extracted from treated leaves using a plant RNA kit (boer, ChongQing, China) according to the manufacturer’s instruction. First strand cDNA was synthesized from 1 µg of total RNA using the PrimerScript^TM^ RT reagent kit with gDNA Eraser (Takara, Japan). The cDNA samples were diluted five times in sterile water before storing at −20 °C. The program for the qRT-PCR was as follows: 5 min at 95 °C, 40 cycles of 5 s at 95 °C and 30 s at 62 °C, 30 s at 95 °C. qRT-PCR was performed in a total volume of 20 μL (0.4 μL each primer (10 μM), 4 μL cDNA, 10 μL SYBR Green Master mix (2×), and 5.2 μL ddH_2_O) in three technological replicates on a QTOWER Real Time PCR System (Analytik Jena, Germany), and the relative expression of the genes was calculated using the 2^−ΔCt^ method [41]. *GhACTIN* was used as the internal standard.

### 4.8. Statistical Analysis

Data for quantification analyses were presented as mean ± standard deviation (SD) with three biological replicates. GraphPad Prism 8.0.2 (https://www.graphpad.com, accessed on 19 September 2022) was utilized to perform significance tests [42]. All statistical tests were two-tailed Student’s *t*-tests at the 0.05 and 0.01 levels, and the *p*-values are given by the symbols * (*p* < 0.05) and ** (*p* < 0.01), respectively.

## 5. Conclusions

In this study, 173 CNGC genes were identified from two diploid and five tetraploid *Gossypium* species. The domain organization, chromosomal location, evolutionary relationships, promoter cis-elements and expression profiles of the *Gossypium* CNGC gene family were then thoroughly examined. The collinearity data demonstrated that four gene losses and three simple translocations occurred during the evolution of Gossypium. Furthermore, the expression of 14 CNGC genes varied significantly after being treated with different hormones and H_2_O_2_. The findings in this study will lay a foundation for unraveling the molecular mechanism of cotton plants’ response to hormonal and exogenous changes.

## Figures and Tables

**Figure 1 ijms-24-04617-f001:**
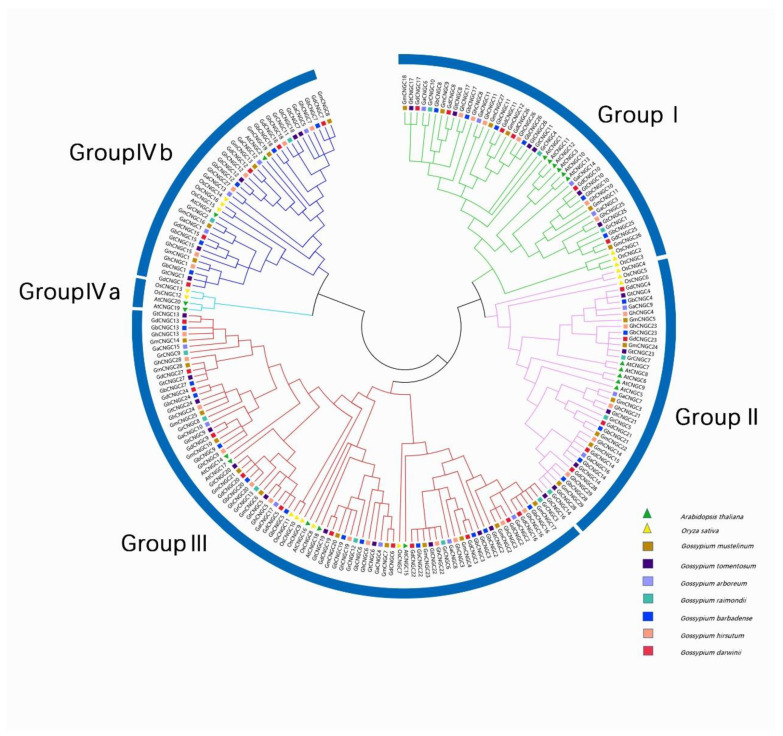
Phylogenetic tree of 9 species including *Arabidopsis thaliana*, *Oryza sativa*, *G. arboreum*, *G. barbadense*, *G. darwinii*, *G. hirsutum*, *G. raimondii*, *G. tomentosum*, *G. mustelinum*.

**Figure 2 ijms-24-04617-f002:**
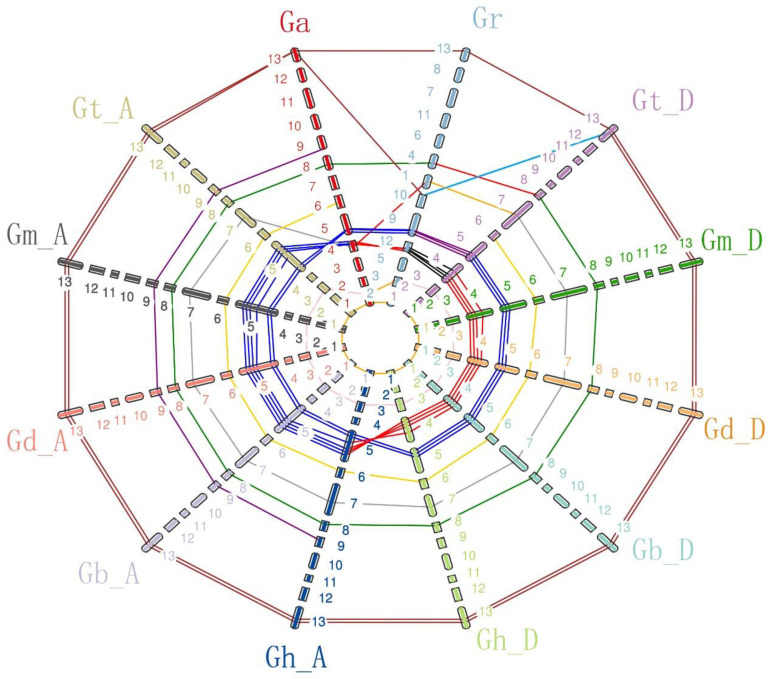
Analysis of *Gossypium* CNGCs for synteny. Ga: *G. arboreum*, Gr: *G. raimondii*, Gt_D: *G. tomentosum* D_t_ subgenome, Gm_D: *G. mustelinum* D_t_ subgenome, Gd_D: *G. darwinii* D_t_ subgenome, Gb_D: *G. barbadense* D_t_ subgenome, Gh_D: *G. hirsutum* D_t_ subgenome, Gh_A: *G. hirsutum* A_t_ subgenome, Gb_A: *G. barbadense* A_t_ subgenome, Gd_A: *G. darwinii* A_t_ subgenome, Gm_A: *G. mustelinum* D_t_ subgenome, Gt_A: *G. tomentosum* D_t_ subgenome. 1–13 represent Chromosome1 to Chromosome13, and different colors represent different chromosomes.

**Figure 3 ijms-24-04617-f003:**
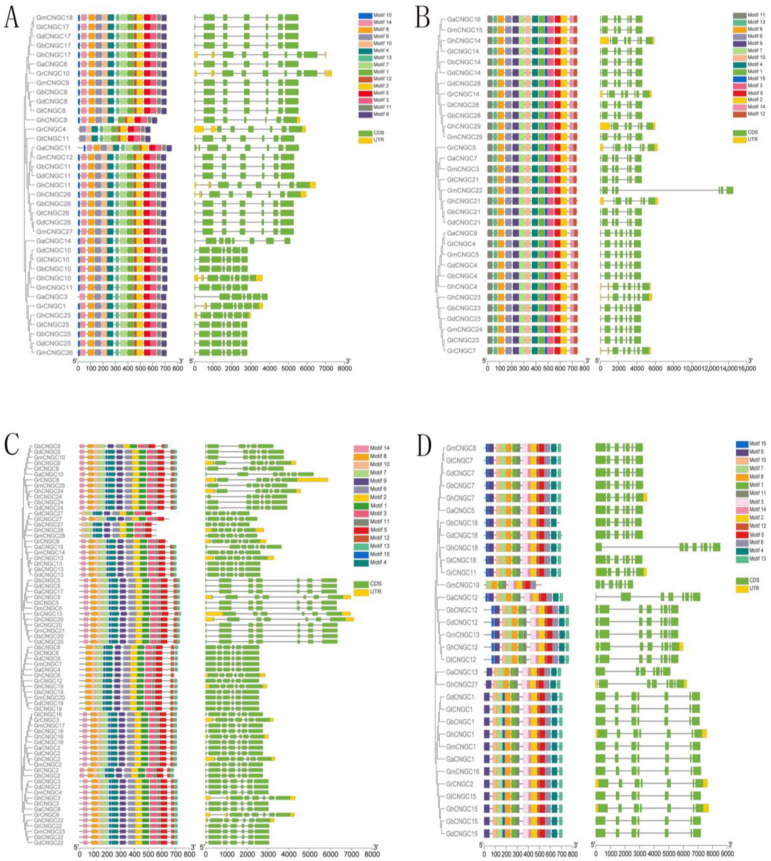
Phylogenetically aligned conserved motifs and gene structure analysis of CNGC genes in seven *Gossypium* species. (**A**) Group I; (**B**) group II; (**C**) group III; (**D**) group IVb.

**Figure 4 ijms-24-04617-f004:**
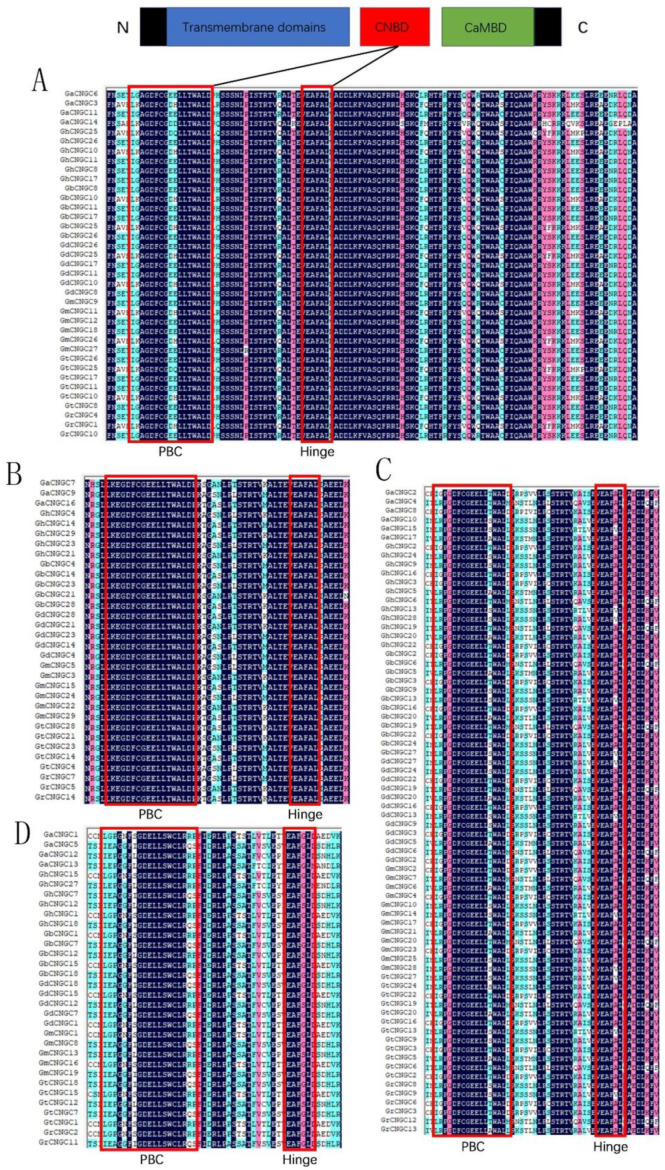
Multiple alignments of CNGC proteins in cotton. (**A**) Group I; (**B**) group II; (**C**) group III; (**D**) group IVb.

**Figure 5 ijms-24-04617-f005:**
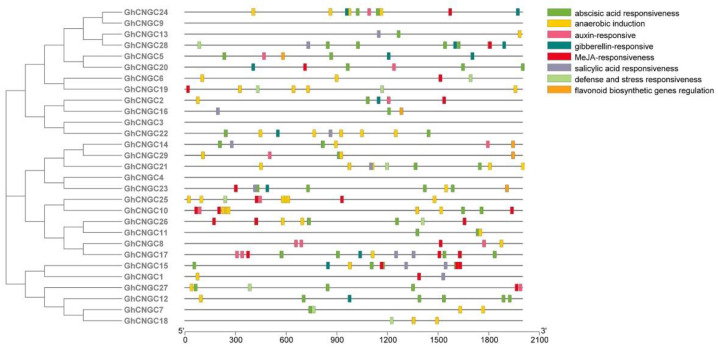
Cis-acting element analysis of GhCNGC genes.

**Figure 6 ijms-24-04617-f006:**
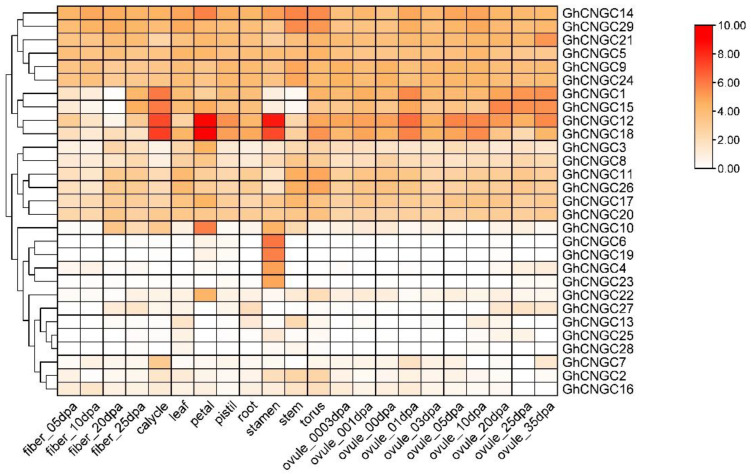
Expression patterns of *GhCNGC* genes: a heatmap of *GhCNGC* genes in fiber developmental stages, calycle, leaf, petal, pistil, root, stamen, stem, torus and ovule developmental stages.

**Figure 7 ijms-24-04617-f007:**
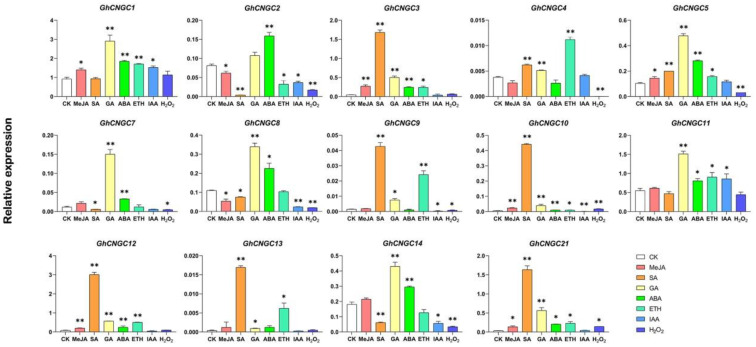
The relative expression levels of selected CNGC genes under control condition (water), abscisic acid (ABA, 100 μM), salicylic acid (SA, 1 mM), methyl jasmonate (MeJA, 100 μM), auxin (IAA, 5 μM), gibberellin (GA 0.5 μM), ethylene (ETH, 200 μM) and H_2_O_2_ (100 mM). All statistical tests were two-tailed Student’s *t*-tests at the 0.05 and 0.01 levels, and the *p*-values are given by the symbols * (*p* < 0.05) and ** (*p* < 0.01), respectively.

## Data Availability

The genome data of the *Gossypium* and RNA-Seq data can be found in the NCBI (http://www.ncbi.nlm.nih.gov/).

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
