# Peer review of "Genome-Wide Identification, Characterization and Experimental Expression Analysis of CNGC Gene Family in Gossypium"

_ijms, 2023, doi:10.3390/ijms24054617_

Round 1

Reviewer 1 Report

please see in the attached file

Author Response

Reviewer #1

Cyclic nucleotide-gated calcium ion channels (CNGCs) play important roles in plant survival and environmental response, while the function of CNGC family in Gossypium is still not well known. In this work, the authors identified 173 genes from two diploid and five tetraploid Gossypium. Phylogenetic analysis showed that these CNGC genes were classified into 4 groups and conservative in seven cottons, four genes loss and three simple translocations were also detected. The various cis-acting regulatory elements in the upstream sequences might be responsible for multiple stimuli. 14 CNGC genes showed significant expression differences after treating with various hormones. This research disclosed related functions of the CNGC gene family in cotton. There are some issues need to be addressed before consideration:

In abstract, “in seven cotton” should be “in seven cottons”.

Response: Thanks for your advice. We had modified this error in the revised manuscript.

In page 2, line 63, the full names of “BLASTp, hmmer, pfam database, SMART and InterPro” need to be presented, moreover, the related references need to be cited.

Response: Thanks for your comment. The detail information such as full names, website and reference of these tools/databases were descripted in the “Materials and methods 4.2 Gene identification and characterization analysis” section. To ensure the results sections are concise and easily understood, we polished sentences related to this comment (Line 68 - 70).

In page 2, line 73 “The theoretical pI of most CNGC proteins was larger than eight”, the “larger” should be “greater”, the full name of “pI” need to be presented as “isoelectric point”, the related references also need to be cited.

Response: We sincerely thank the reviewer for careful reading. As suggested by the reviewer, we have revised the “larger” into “greater”. Besides, the information of “pI” was supplemented in the corresponding section in the Materials and methods 4.3.

In page 11, line 283, The authors mentioned “ethylene (ETH, 200μM) were applied to sprayed...”, the solvent of ethylene solution must be presented.

Response: Thanks for your comment. The solvents of all treating solutions were provided this information in the revised manuscript (section 4.1 Plant growth and treatments).

The conclusion part is a bit simple; the main results need to be presented and discussed; the authors please re-organize this part.

Response: Thanks for your comment. The “Conclusion” was organized as a single section in the revised manuscript, and the corresponding contents were revised and improved following the comments from both reviewers.

Reviewer 2 Report

The manuscript “ijms-2125165” entitled “Genome-wide identification, characterization and experimental expression analysis of CNGC gene family in Gossypium” by Chen et al. deals with an interesting subject focused on the function of the CNGC family in Gossypium. In this study, 173 genes were identified from two diploid and five tetraploid Gossypium. After phylogenetic analysis, these CNGC genes were classified into 4 groups. The collinearity results show that CNGC genes are conservative in seven cotton. Four genes loss and three simple translocations were detected, which is beneficial to analyze the evolution of CNGCs in Gossypium. The various cis-acting regulatory elements in the upstream sequences revealed the possible function in responding to multiple stimuli, including hormonal, and abiotic stresses. In addition, 14 CNGC genes showed significant expression differences after treating with various hormones.

For publication in the “IJMS” journal, the topic and content are appropriate. The subject of the study is interesting with high scientific and practical importance. The introduction is in accordance with the subject and correctly presented. The methodology of the study was clearly presented, and appropriate to the proposed objectives. The obtained results have been analyzed and interpreted in accordance with the current methodology. The discussions are appropriate, in the context of the results, and were conducted compared to other studies in the field. The scientific literature, to which the reporting was made, is recent and representative in the field. However, the review of the article revealed some issues, which were noted in the article and listed below:

Materials and Methods: Statistical analysis sub-section is missing. Please write this sub-section including the experimental layout they used in their study and the statistical software package used for the analysis. In addition, please refer to the statistical test used to determine the differences between the parameters at all treatments (e.g. Figure 7 - the relative expression levels of selected CNGC genes under several oxidative stress and hormones levels).

Discussion: The discussion section must be enhanced. The authors should further refer to previous studies concerning the function of the CNGC family in other plant species.

Lines 238-239: Scientific names must be italicized.

Conclusions section: The conclusions section is missing. This should be summarized in one paragraph and restate the thesis, summarize the key supporting ideas discussed throughout the work, and offer the final impression of the central idea.

Finally, the reviewer recommends the authors carefully revise the paper format and back matter section (author contributions, funding, etc.) and be consistent with the formatting of references and cross-references. This has to be standardized across the paper. For more details, please see “Instructions for authors” 

Thank you for your consideration.

Author Response

Reviewer #2

The manuscript “ijms-2125165” entitled “Genome-wide identification, characterization and experimental expression analysis of CNGC gene family in Gossypium” by Chen et al. deals with an interesting subject focused on the function of the CNGC family in Gossypium. In this study, 173 genes were identified from two diploid and five tetraploid Gossypium. After phylogenetic analysis, these CNGC genes were classified into 4 groups. The collinearity results show that CNGC genes are conservative in seven cotton. Four genes loss and three simple translocations were detected, which is beneficial to analyze the evolution of CNGCs in Gossypium. The various cis-acting regulatory elements in the upstream sequences revealed the possible function in responding to multiple stimuli, including hormonal, and abiotic stresses. In addition, 14 CNGC genes showed significant expression differences after treating with various hormones.

For publication in the “IJMS” journal, the topic and content are appropriate. The subject of the study is interesting with high scientific and practical importance. The introduction is in accordance with the subject and correctly presented. The methodology of the study was clearly presented, and appropriate to the proposed objectives. The obtained results have been analyzed and interpreted in accordance with the current methodology. The discussions are appropriate, in the context of the results, and were conducted compared to other studies in the field. The scientific literature, to which the reporting was made, is recent and representative in the field. However, the review of the article revealed some issues, which were noted in the article and listed below:

Materials and Methods: Statistical analysis sub-section is missing. Please write this sub-section including the experimental layout they used in their study and the statistical software package used for the analysis. In addition, please refer to the statistical test used to determine the differences between the parameters at all treatments (e.g. Figure 7 - the relative expression levels of selected CNGC genes under several oxidative stress and hormones levels).

 Response: Thanks for your advice. The Statistical Analysis section was supplied in the Materials and methods 4.8, and the analysis for statistical test was also provided in this section.

Discussion: The discussion section must be enhanced. The authors should further refer to previous studies concerning the function of the CNGC family in other plant species.

 Response: Thanks for your comment. The “Discussion” section was improved by supplementing new contents according to this comment. Studies of CNGCs in plant species such as rice, tobacco, and Chinese cabbage were taken as examples to further discuss the characteristics and functions of CNGC family among plants.

Lines 238-239: Scientific names must be italicized.

 Response: Thanks for your advice. We had revised these errors in revised manuscript.

Conclusions section: The conclusions section is missing. This should be summarized in one paragraph and restate the thesis, summarize the key supporting ideas discussed throughout the work, and offer the final impression of the central idea.

  Response: Thanks for your comment. The “Conclusion” was organized as a single section in the revised manuscript, and the corresponding contents were revised and improved following the comments from both reviewers.

Finally, the reviewer recommends the authors carefully revise the paper format and back matter section (author contributions, funding, etc.) and be consistent with the formatting of references and cross-references. This has to be standardized across the paper. For more details, please see “Instructions for authors” 

Response: We are sorry for our negligence. We had modified these mistakes in revised manuscript.

Round 2

Reviewer 2 Report

The text has been corrected according to my suggestions. Responses to comments are satisfactory. I recommend the manuscript be published in the IJMS journal.